# Research Progress on Chemical Constituents and Pharmacological Activities of *Menispermi Rhizoma*

**DOI:** 10.3390/molecules28062701

**Published:** 2023-03-16

**Authors:** Xuan Zhai, Kangmin Wang, Xingyi Gao, Bin Yan

**Affiliations:** 1College of Pharmacy, Shandong University of Traditional Chinese Medicine, Changqing District, Jinan 250355, China; 2College of Traditional Chinese Medicine, Shandong University of Traditional Chinese Medicine, Changqing District, Jinan 250355, China

**Keywords:** *Menispermi Rhizoma*, chemical constituents, pharmacological activities, research progress

## Abstract

*Menispermi Rhizoma*, the rhizome of *Menispermum dauricum* DC., is a traditional Chinese medicine, which has the effect of clearing away heat and detoxification, dispelling wind, and relieving pain. It is often used in the treatment of sore throat, enteritis, dysentery, and rheumatism. The chemical constituents of *M. Rhizoma* mainly include alkaloids, phenolic acids, quinones, cardiotonic glycosides, and so on. Modern pharmacological studies have proved that *M. Rhizoma* has the effects of anti-tumour, anti-inflammation, anti-oxidation, bacteriostasis, cardio-cerebrovascular protection, anti-depression and anti-Alzheimer’s disease. In recent years, the chemical constituents of *M. Rhizoma* have been found continuously, and the pharmacological studies have deepened gradually. This paper reviews the research progress on the chemical composition and pharmacological effects of *M. Rhizoma*, to provide a basis for further research and development of its medicinal value.

## 1. Introduction

*Menispermi Rhizoma* is the dried rhizome of *Menispermum dauricum* DC. It is mainly produced in Northeast China, North China, East China, and Shaanxi. It has the effect of clearing heat and detoxifying, dispelling wind, and relieving pain, and is mainly used for sore throat, pyretic diarrhoea, dysentery, and rheumatic paralysis [1]. Modern research has shown that the various chemical components contained in *M. Rhizoma*, including alkaloids, phenolic acids, quinones, cardiac glycosides, and polysaccharides, have a variety of pharmacological activities, such as anti-tumour, anti-inflammatory, antioxidant, antibacterial, cardiovascular, antidepressant, and anti-Alzheimer’s disease [2,3]. This review reports the research progress on chemical constituents and pharmacological activities of *M. Rhizoma* up to 2023. First, all the chemical components of *M. Rhizoma* that have been discovered so far are listed. The most abundant component of *M. Rhizoma*, the alkaloid, was systematically classified based on their structural characteristics. After that, the pharmacological activities and applications of *M. Rhizoma* are also reported according to the type of clinical disease treated. In summary, this review will provide a theoretical basis for further research and the utilization of *M. Rhizoma* in the future.

## 2. Chemical Composition

So far, more than 100 compounds, including alkaloids, phenolic acids, quinones, cardiac glycosides, polysaccharides, and other chemical components, have been isolated and identified from *M. Rhizoma* [3,4].

### 2.1. Alkaloids

Alkaloids, as the signature components of *M. Rhizoma*, are also the most abundant class of components, with a content of 1.7–2.5% [3,4]. The diversity of alkaloid structures in *M. Rhizoma* is mainly distinguished by their parent nucleus structure, the type and number of substituents, and chiral carbon atoms. The main species include bisbenzylisoquinolines (Table 1: 1–45), apomorphins and oxidized isoapomorphins (Table 1: 46–82), morpholines (Table 1: 83–91), proberberberine and berberine (Table 1: 92–98), and other classes of alkaloids (Table 1: 99–117). Among them, bibenzylisoquinolines, apomorphins, and oxidized isoapomorphins alkaloids are the most distributed among the alkaloid components of *M. Rhizoma*.

#### 2.1.1. Bisbenzylisoquinoline Alkaloids

Bisbenzylisoquinoline alkaloids contain two benzylisoquinolines linked by diphenyl ether, benzyl phenyl ether or biphenyl bonds [24]. The structural skeleton I of bisbenzylisoquinoline alkaloids from *M. Rhizoma* is shown in Figure 1, where the substituents R_1_–R_5_ are often H or CH_3_. Alkaloids 1–36 belong to this category, where a mixture of multiple lipid-soluble alkaloids with alkaloids 1 and 4 as the main components is also known as Phenolic Alkaloids from *Menisphermum dauricum* (PAMD) [25]. There are several isomers in this structure due to the different C_1_-H, C_1′_-H and C_2_-CH_3_, C_2′_-CH_3_ space configurations, such as alkaloids 12 and 13, and the difference between them is the difference in the C_1_-H space configuration. Since the structural changes within the molecules of this class of alkaloids are mainly in the number of aromatic oxygens, the number of ether bonds, the nature of oxygen bridges, the position of carbon-carbon bond initiation on the alkaloid units, and the nature of nitrogen atom substituents, these structural changes are highly likely to produce new structures of bisbenzylisoquinoline alkaloids and new skeletons [10]. The structural skeleton II, also shown in Figure 1, differs from the structural skeleton I by the change in the position of the connection between the two benzylisoquinolines. The C_7′_ position in this structure is connected to the C_11_ or C_12_ position, and the C7 position is often connected to the C11′ or C12′ position by an oxygen bridge or replaced by OH or OCH_3_, as in the C_5_, C_5′_, or C_8′_ positions. All alkaloids 37–44 found in the extract of *M. Rhizoma* have this feature, with alkaloid 37 and alkaloid 44 being trace alkaloids obtained from *M. Rhizoma* for the first time. Compared with the typical bisbenzylisoquinoline structural skeleton I of *M. Rhizoma*, alkaloid 45 has a broken bond between the C_1′_ and C_9′_ positions and undergoes carbonylation to become a ring-cleaving bisbenzylisoquinoline structure. The bisbenzylisoquinoline alkaloids that have been identified in *M. Rhizoma* are shown in Figure 2 and Figure 3. In addition to the above bisbenzylisoquinoline alkaloids, Li et al. [7] identified four other bisbenzyltetrahydroisoquinoline alkaloids from *M. Rhizoma* with the help of the UPLC-Q-TOF-MS/MS technique, namely, *N*-demethylepiphylline, 2-demethylepiphylline, 5-hydroxylepiphylline, and tamsulosin.

#### 2.1.2. Apomorphines and Oxidized Isoporphine Alkaloids

The alkaloids are based on a tetracyclic aromatic backbone formed by the oxidative coupling of the phenol of the benzylisoquinoline precursor [26]; the characteristic tetracyclic system (rings A–D), in which ring B often contains a nitrogen atom [24], is shown in Figure 4. In addition, the oxidized isoporphine alkaloids also have a four-ring system, which differs from the apomorphine alkaloids in the oxidation of the ring C methylene and the position of the linkage of ring D. The structural skeleton III is shown in Figure 4. The alkaloids 63–82 found in *M. Rhizoma* belong to this group of alkaloids, where alkaloid 82 differs from other oxidized iso-apomorphine alkaloids in *M. Rhizoma* by the breakage and oxidation of ring A to the carbonyl group and the reduction of ring C carbonyl group to the hydroxyl group. It should be noted that alkaloids 69, 70, and 72 are identified as the new compounds of oxidized isoporphine alkaloids [16,20]. Apomorphine alkaloids and oxidized isoporphine alkaloids have been identified in *M. Rhizoma* are shown in Figure 5 and Figure 6, respectively.

#### 2.1.3. Morphine Alkaloids, Proberberberine, Berberine, and Other Alkaloids

The morphine alkaloids, proberberberine, and berberine alkaloids that have been extracted and isolated from *M. Rhizoma* are detailed in Figure 7 and Figure 8. Among them, alkaloids 87–91, which are chlorinated alkaloids with a new backbone, have been discovered in *M. Rhizoma* in recent years. In addition, the berberine alkaloid 98, derived from the *n*-butanol part of the 50% ethanol extract of *M. Rhizoma*, was found for the first time in the genus Batrachochia [14]. In addition to the above alkaloids, other classes of alkaloids currently found in *M. Rhizoma* are shown in Figure 9. Wei et al. [13] isolated apomorphine–benzylisoquinoline alkaloids 99 and 100 from *M. Rhizoma*, and other studies obtained simple isoquinoline alkaloids (alkaloids 101–106) and monobenzylisoquinoline alkaloids (alkaloids 107–117) from *M. Rhizoma*. Among them, Chen et al. [23] identified three newly discovered alkaloids 115–117 obtained in the dichlorinated carbon part of the 95% ethanol extract of *M. Rhizoma* as simple isoquinoline alkaloids by the nuclear magnetic resonance technique.

### 2.2. Other Components

In addition to alkaloids, *M. Rhizoma* contains volatile components, polysaccharides, quinones, cardiac glycosides, lactones, saponins, tannins, proteins, and resins, among other chemical constituents [21]. In recent years, some components isolated for the first time from *Menispermaceae* or *Menispermum* Linn. or *M. Rhizoma* have been discovered, greatly enriching the chemical composition of *M. Rhizoma*. Compounds **1** and **2** were isolated from the dichloromethane part of 50% ethanol extract of *M. Rhizoma* by Li et al. [14], where compound **1** was obtained for the first time from *Menispermum* Linn. Compounds **7**–**9**, which are nephrotoxic, were first isolated from *M. Rhizoma* [4,5]; compounds **14**–**17**, **19**, **22** were first isolated from *Menispermaceae*; and compounds **12**–**23** were first isolated from *Menispermum* Linn. [27]. In addition, Ren et al. [28] analysed the composition of the fatty oil of *M. Rhizoma* by GC-MS; the specific components are shown in Table 2, compounds **24**–**55**. In addition to the above components, Lin et al. [29,30] also obtained one water-soluble polysaccharide WMDP with a triple-helix structure and two acidic polysaccharides MDP-A1 and MDP-A2 from *M. Rhizoma*.

## 3. Pharmacological Activities

A diverse and structurally complex group of alkaloids is one of the characteristics of the chemical composition of *M. Rhizoma*. In recent years, many scholars have conducted extensive research on the biological activities of alkaloids in *M. Rhizoma*, while the discoveries of other components of *M. Rhizoma* and their activities have also greatly enriched the material basis of the medicinal effects of *M. Rhizoma*. It has been found to play an important role in anti-tumour, anti-inflammatory, antioxidant, antibacterial, cardio-protective, anti-depressant, and anti-Alzheimer’s disease.

### 3.1. Anti-Tumour Effect

A network pharmacology-based study investigating the antihepatocarcinogenic mechanism of isoquinoline alkaloids concluded that tetrandrine could exert antihepatocarcinogenic effects by inducing cellular autophagy, inhibiting tumour cell invasion and metastasis, and enhancing radiosensitivity [31]. The results of another tumour cytotoxic activity test showed that thalifortine, cycleapeltine, sutchuenenine, and menisperine had good inhibitory activity on the proliferation of human hepatoma HepG2 cells [10]. PAMD, with dauricine and daurisoline as the main components, has been used as a broad-spectrum anti-tumour active ingredient in recent years. Recent studies have shown that PAMD can down-regulate the expression of Shh, Ptch1, Smo and Gli1, key loci of the Hedgehog signalling pathway, and inhibit the growth of tumour cells, thus achieving anti-tumour effects [32]. In addition, daurisoline was able to induce apoptosis in human hepatoma cells HepG2 and Hep3B, promote Hep3B cell necrosis, and inhibit the migration ability of hepatocellular carcinoma cells [33], while dauricine can exert anti-tumour effects by inhibiting the proliferation of SW1900 and BxPC-3 pancreatic cancer cells [34,35], Hela cervical cancer cells [36], Huh7 liver cancer cells [37], Eca-109 esophageal cancer cells [38], A375 and A2058 melanoma cells [39], kidney cancer cells [40], colon cancer cells [41], EJ-1 and 5637 bladder cancer cells [42], and CNE-2 nasopharyngeal cancer cells [43]. All the alkaloids mentioned above belong to the bisbenzylisoquinoline group of alkaloids, indicating the indispensable role of this group of alkaloids in the anti-tumour activity exerted by *M. Rhizoma*. In addition, studies have shown that morphine alkaloids acutumine are generally effective in inhibiting SMMC-7721 human liver cancer cells, MCF-7 human breast cancer cells, A549 human lung cancer cells, SW-480 human intestinal cancer cells, and HL-60 human leukemia cells [22]. A water-soluble polysaccharide WMDP with a triple-helix structure and two acidic polysaccharides MDP-A1 and MDP-A2 extracted from *M. Rhizoma* by Lin et al. [29,30] could significantly inhibit the proliferation of SKOV3 human ovarian cancer cells and effectively induce the apoptosis of SKOV3 cells, which indicated the potential application of the above polysaccharides as natural anti-tumour drugs and provided a scientific basis for the in-depth study of the active components of *M. Rhizoma* that exert anti-tumour effects.

### 3.2. Anti-Inflammatory Effect

Ulcerative colitis (UC)—characterized by abdominal pain; diarrhoea; and mucous, bloody stools as the main clinical manifestations—has a high recurrence rate and is difficult to cure [44]. Studies have shown that bisbenzylisoquinoline alkaloids dauricine, daurinoline, dauricinoline, daurisoline and tetrandrine, and morphine alkaloids sinomenine and acutumine can reduce the expression of MPO and COX-2, down-regulate the expression of NF-κB/TLR4 mRNA and significantly reduce the levels of TNF-α, IL-1β, and IL-6 in mouse colon tissues to varying degrees, suggesting that *M. Rhizoma* has anti-inflammatory and promotes the repair of damaged tissues in the colon [6]. Similar studies have demonstrated that in addition to lowering the serum levels of IL-6, MMP and VEGF levels are also down-regulated in the treatment of UC by northern bean root monosodium glutamate [45]. Network pharmacological studies on the mechanism of action of *M. Rhizoma* in the treatment of UC have demonstrated that *M. Rhizoma* may intervene in digestive and immune systems by participating in biological processes such as the regulation of cell proliferation and apoptosis; signalling; transcriptional regulation and drug response; and modulating pathways such as TNF, PI3K-Akt, T-cell receptor signalling, etc. The components involved in the drug-active ingredient-target network include morphine alkaloids dauricumine and acutuminine, and bisbenzylisoquinoline alkaloids stepharine, bianfugecine, and stepholidine [46]. In addition to a good effect of the active constituents in *M. Rhizoma* for the treatment of UC, the bisbenzylisoquinoline alkaloids (+)-1,3,4-dehydrocepharanthine, cissampentine A, cissampentine B and (−)-pseudocurine, apocynine alkaloid 6-acetyl-5,6-dihydro-1,2-dimethoxy-4H-dibenzo[de,g]-quinoline, oxidized isoapocynine alkaloids oxoisoaporphine B, menisoxoisoaporphine A, and daurioxoisoporphine B in *M. Rhizoma* showed good inhibitory activity against the release of NO from rat macrophages in the LPS-induced anti-inflammatory activity assay [11,17]. In another study, it was demonstrated that the total alkaloids of *M. Rhizoma* inhibit ovalbumin-induced airway inflammation in mice with asthma by reducing the concentrations of interleukin 4, 5, and 13, down-regulating the levels of TNF-α and eotax in bronchoalveolar lavage fluid, and inhibiting the increase in serum levels of total immunoglobulin E and ovalbumin-specific immunoglobulin E. The results of this experiment suggest that the total alkaloids of *M. Rhizoma* can inhibit ovalbumin-induced airway inflammation in mice by modulating T-helper 2 responses and chemokine levels, suggesting that the total alkaloids of *M. Rhizoma* may be potential anti-asthmatic agents [47]. In summary, the alkaloids of *M. Rhizoma*, especially bisbenzylisoquinoline and morphinane alkaloids, are the significant pharmacological bases for the anti-inflammatory activity of *M. Rhizoma*. In addition, arachidic acid obtained from the methanolic extract of *M. Rhizoma* by Ren et al. [27] showed a strong inhibition of NO and IL-6 release from RAW 264.7 cells, suggesting that it has good anti-inflammatory activity in vitro, which provides a scientific and theoretical basis for the subsequent search for new anti-inflammatory components of *M. Rhizoma*.

### 3.3. Antioxidant Effect

DPPH radicals are commonly used for in vitro antioxidant activity evaluation, and the stronger the scavenging ability of DPPH radicals, the stronger the antioxidant capacity. The scavenging rate of oxidized apomorphine alkaloids dauriporphine and menisporphine on DPPH radicals was similar to that of the positive reference drug vitamin C, both above 90%, providing an experimental basis for the development of antioxidant drugs from the alkaloids of *M. Rhizoma* [48]. In addition, Ren et al. [28] found that the fatty oil of *M. Rhizoma* also had a better scavenging ability for DPPH radicals with a scavenging rate of 70.1%; thus, it was speculated that the long-chain unsaturated fatty acid methyl esters such as methyllinoleate and methyloleate, which were more abundant in the fatty oil of *M. Rhizoma* identified by GC-MS, might be related to the antioxidant activity of the fatty oil of *M. Rhizoma*.

### 3.4. Antibacterial Effect

The alkaloid components of *M. Rhizoma* were reported to have inhibitory effects on a variety of respiratory and intestinal bacteria, with the most significant inhibitory effect on dauricine, with an inhibition rate of 83.33%, and the best inhibitory effect on *S. pneumoniae* [49]. Clinical studies further confirmed that dauricine also inhibited *E. coli*, *S. aureus*, and *B. subtilis* to different degrees, and the inhibitory effect was: *B. subtilis* > *S. aureus* > *E. coli* [50].

### 3.5. Cardio-Protective Effect

Dauricine is often used as a clinical treatment for hypertension and cardiac arrhythmias. Its antihypertensive effect was reported to be related to the antagonism of Ca^2+^ channels, and its antiarrhythmic mechanism of action is similar to that of the class III antiarrhythmic drug amiodarone: it mildly inhibits Ca^2+^-ATPase activity, decreases sarcoplasmic reticulum calcium uptake and has the effect of inhibiting Na^+^ inward flow, Ca^2+^ inward flow, and K^+^ outward flow, especially blocking K^+^ outward flow [51]. Ischemic cerebrovascular diseases such as ischemic stroke and stroke often lead to damage and disruption of the blood–brain barrier and accompanying cerebral edema. Establishing animal models of focal cerebral ischemia and reperfusion injury in the brain of rats is one of the most critical tools for studying the pathophysiological mechanisms of these cardiovascular diseases [52]. Zhang et al. [53] found that PAMD reduced the water content of brain tissue in this animal model of injury and reduced the permeability of blood–brain barrier, which was associated with the upregulation of the p-NR1 expression by PAMD and thus reduced the incidence of NMDAR activation. Additional studies have demonstrated that dauricine, one of the main components of PAMD, protects against ischaemia-reperfused brain tissue damage by inhibiting the expression of P-glycoprotein in brain tissue and achieving reverse retention of this alkaloid in brain tissue [52]. Other five oxidized isoporphine alkaloids showed good anti-myocardial ischaemic activity, with menisporphine, dauriporphinoline, and oxoisoaporphine A exhibiting a good anti-myocardial ischaemic effect by effectively increasing the survival of cardiomyocytes damaged by glyoxylate deprivation [20].

### 3.6. Anti-Hypoxic Effect

Shao et al. [4] found in the study of the anti-hypoxic activity of the chemical constituents of *M. Rhizoma* that the protective effect of bisbenzylisoquinoline and morphine alkaloids on hypoxia-injured EA.hy926 vascular endothelial cells were more obvious. The more abundant bisbenzylisoquinoline alkaloid daurisoline in *M. Rhizoma* showed the strongest anti-hypoxic activity, followed by morphine alkaloids acutumine and acutuminine. The above studies provide the material basis for the better anti-hypoxic activity of *M. Rhizoma*.

### 3.7. Anti-Depressant Effect

Depressed patients tend to have decreased levels of 5-hydroxytryptamine (5-HT), and certain genetic polymorphisms in 5-HT metabolism and transporters are associated with depression [54]. Studies have confirmed that 5-HT can be catabolized and deaminated by residual MAO-A in the capillaries [54]. Several synthetic dihydro and oxo-isoporphine derivatives were evaluated by in vitro experiments, and the results showed that all dihydro and oxo-isoporphine derivatives tested were selective MAO-A inhibitors, with the most representative and potent in vitro MAO-A inhibitor being 5-methoxyoxoisoaporphine (OXO4), an oxo-isoporphine derivative synthesized from *M. Rhizoma* [55]. The compulsive swimming trial added to the evidence that OXO4 requires a smaller dose for the same duration of action to achieve the same antidepressant effect compared to classical antidepressants [55]. Based on these facts, the oxidized isoporphine alkaloids in *M. Rhizoma* are expected to be developed as more efficient antidepressants.

### 3.8. Anti-Alzheimer’s Disease Effect

One of the primary pathogeneses of Alzheimer’s disease (AD) that is now widely recognized is its association with impairment in cholinergic transmission processes, where patients with low acetylcholine levels and reduced function in the brain experience significant cognitive impairment. The results of in vitro enzyme activity experiments showed that the alkaloids in *M. Rhizoma* inhibited acetylcholinesterase (AChE), with the monobenzylisoquinoline alkaloid pecrassipine B having the most significant inhibitory effect on AChE, followed by that of the bisbenzylisoquinoline alkaloid daurisoline, the morpholino alkaloid acutumine, the simpleisoquinoline alkaloid thalifoline, pycnarrhine, and amurolin; the inhibition effect of the simpleisoquinoline alkaloid corypalline is weaker in comparison. The molecular docking results showed that the strength of AChE inhibition by pecrassipine B and corypallinewas was related to their respective molecular structures and the degree of AChE binding [22]. Neurotoxic amyloid β-protein (Aβ) is a major component of neuroinflammatory plaques. Related studies have demonstrated that Aβ exerts neurotoxic effects and induces neuronal apoptosis by increasing the expression level of the pro-apoptotic gene Bax, decreasing that of the anti-apoptotic gene Bcl-2 [56]. It has also been shown that dauricine can significantly reduce the levels of IL-1β, IL-6, RAGE, and NF-κBp65 in the hippocampus of mice and decrease Aβ accumulation, thus delaying the course of AD [57]. Two other research results provided new ideas for the treatment of AD with *M. Rhizoma*: Wang [58] used the Nrf2/Keap1 antioxidant pathway to verify that dauricine could significantly increase the expression level of Nrf2, a key antioxidant factor, and then used the antioxidant effect of dauricine to repair damaged cells using Aβ aggregation as a therapeutic target; the results demonstrated that dauricine could be brain-targeted for the treatment of AD. A similar study in which dauricine was applied to an AD transgenic cell model resulted in a gradual increase or decrease in cell survival and MDA content and a gradual decrease in COX-2 protein expression in the AD transgenic model, demonstrating the protective effect of dauricine against oxidative damage in this model [59]. In summary, it is reasonable to assume that the isoquinoline alkaloids in *M. Rhizoma* are potentially promising for the prevention and treatment of AD.

### 3.9. Toxicity

*M. Rhizoma* is slightly toxic and clinical application is accompanied by adverse effects such as nausea, vomiting, loss of appetite, dyspepsia, bloating, and diarrhoea. Studies have shown that the acute toxicity of the alcoholic fraction of *M. Rhizoma* is greater than that of the aqueous fraction [60], and the total alkaloids of *M. Rhizoma* are the major alcohol-soluble components, thus verifying the studies on the chemical composition of *M. Rhizoma* reported in the literature [61]. This suggests that the alkaloids contained in *M. Rhizoma* are the main material basis for its toxicity. The toxic effects of the aqueous and alcoholic fractions of *M. Rhizoma* manifested as acute or chronic hepatotoxic injury with significant changes in serum ALT, AST, and hepatic body ratios [62]. Similar studies have further confirmed that the water and alcohol extraction of *M. Rhizoma* can increase MDA content in liver tissue and decrease SOD activity; this confirms, at the intrahepatic substance level, that the mechanism of hepatotoxic injury caused by *M. Rhizoma* is related to the induction of lipid peroxidation and reduction of its own redox capacity after causing oxidative stress in the body, as well as to the NO-mediated damage pathway [62].

## 4. Conclusions

With a wide distribution range and abundant medicinal resources, *M. Rhizoma* has a long history of medicinal use. In recent years, domestic and foreign scholars have conducted extensive and in-depth studies on the chemical components of *M. Rhizoma*, especially alkaloid components, and up to now, more than 150 chemical components, including 117 alkaloids, have been identified from *M. Rhizoma*. Among the alkaloid components of *M. Rhizome*, bisbenzylisoquinolines are predominant, and apomorphines and oxidized isoporphines are the next most abundant; the effects of *M. Rhizoma* in anti-tumour, antioxidant, anti-inflammatory, antibacterial, cardiovascular and cerebrovascular protection, anti-depressant, and anti-Alzheimer’s disease have been gradually confirmed and applied in the treatment of clinical diseases. The above research results have carried forward the modernization of traditional medicines. To summarize the current results, three points need attention for further research on *M. Rhizoma*: Firstly, in the pharmacological research on the alkaloid components of *M. Rhizoma*, most of the studies focused on the PAMD and relatively few studies on other alkaloids. Compared with the current research, the pharmacological studies of other chemical components isolated from *M. Rhizoma* are relatively lacking. Thirdly, the pharmacological study on *M. Rhizoma* can also be combined with the knowledge of molecular biology, proteomics, metabolomics, and other disciplines to investigate further the targets, mechanisms, and metabolic patterns of its effects. This paper reviews the progress of research on the chemical composition and pharmacological effects of *M. Rhizoma* in recent years and provides a basis for the further development and utilization of *M. Rhizoma*.

## Figures and Tables

**Figure 1 molecules-28-02701-f001:**
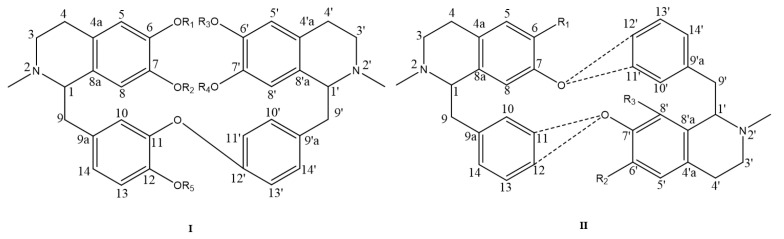
Structural skeletons of bisbenzylisoquinoline alkaloids in *M. Rhizoma*.

**Figure 2 molecules-28-02701-f002:**
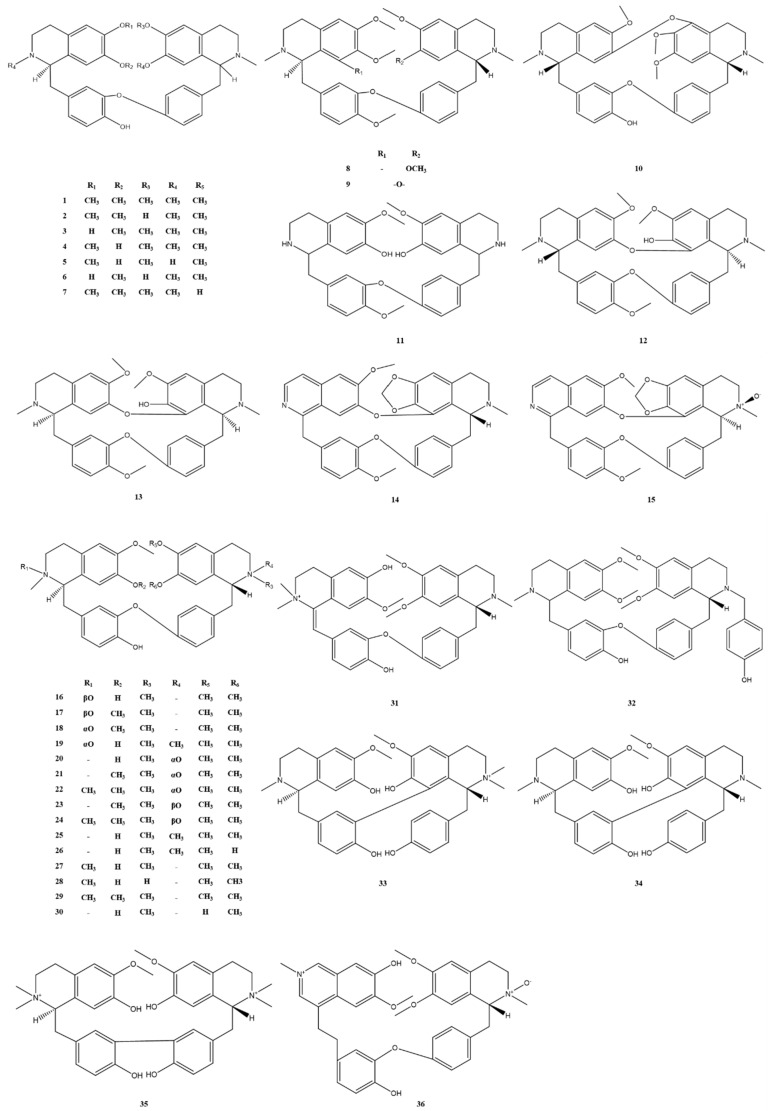
Bisbenzylisoquinoline alkaloids (structural skeleton I) in *M. Rhizoma*.

**Figure 3 molecules-28-02701-f003:**
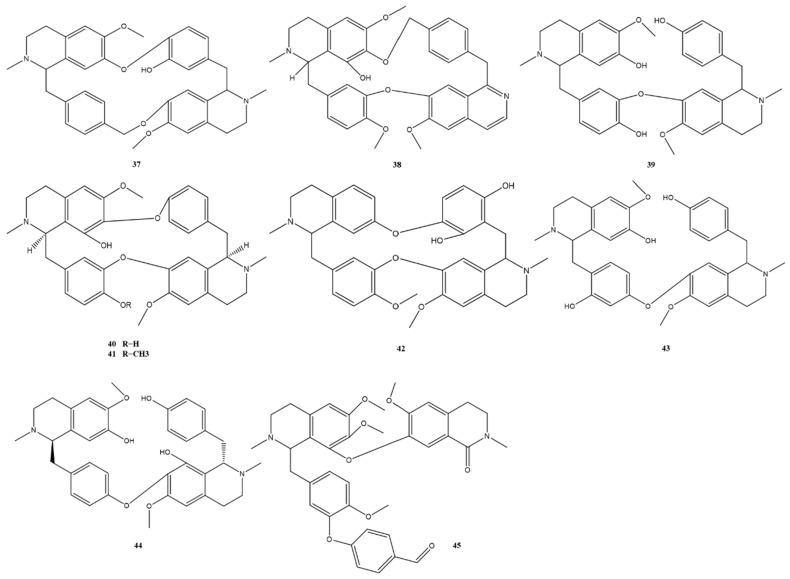
Bisbenzylisoquinoline alkaloids (structural skeleton II) in *M. Rhizoma*.

**Figure 4 molecules-28-02701-f004:**
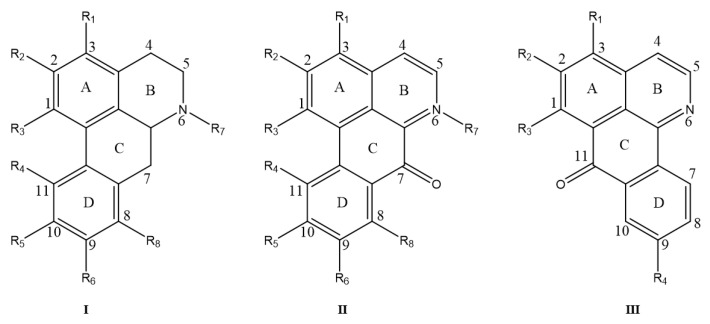
Structural skeletons of apomorphines and oxidized isoapomorphine alkaloids in *M. Rhizoma*.

**Figure 5 molecules-28-02701-f005:**
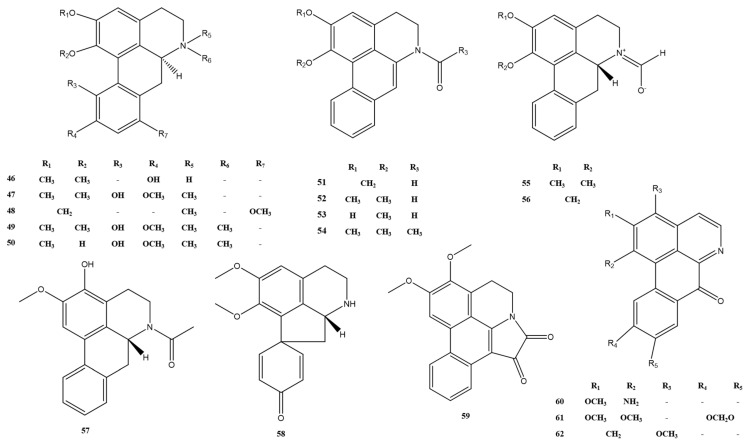
Apomorphine alkaloids in *M. Rhizoma*.

**Figure 6 molecules-28-02701-f006:**
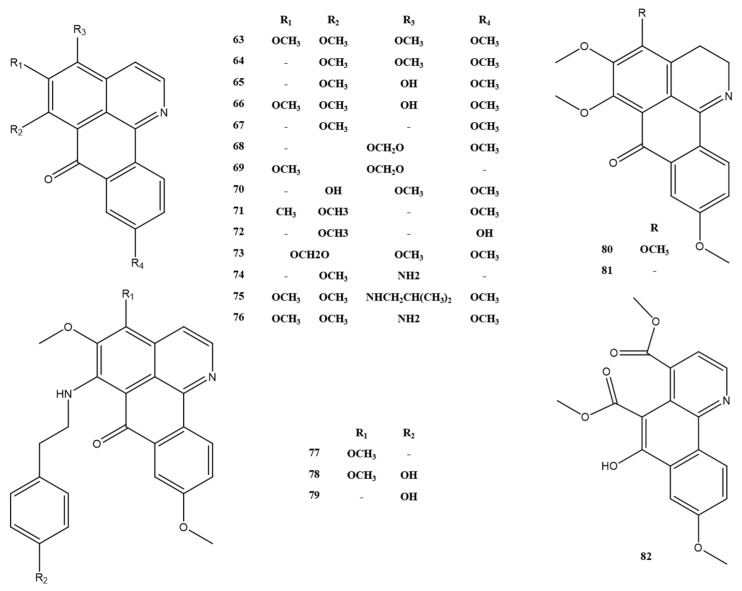
Oxidized isoporphine alkaloids in *M. Rhizoma*.

**Figure 7 molecules-28-02701-f007:**
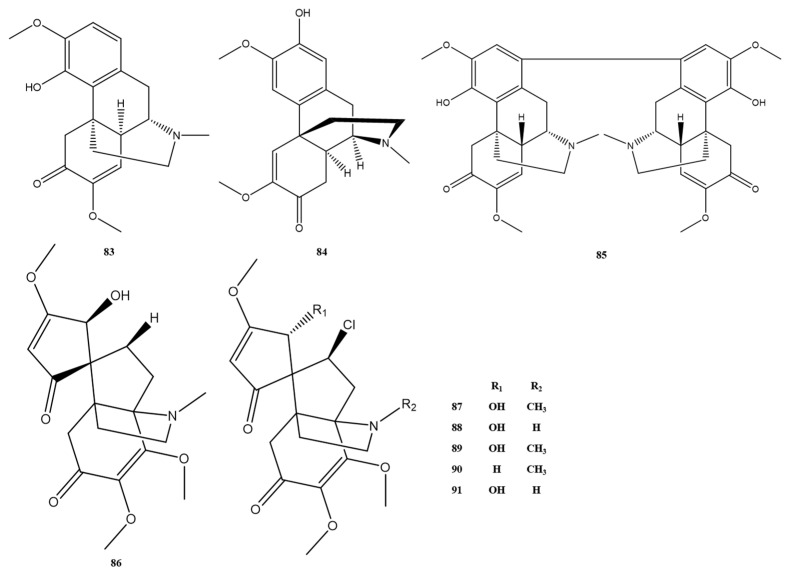
Morphine alkaloids in *M. Rhizoma*.

**Figure 8 molecules-28-02701-f008:**
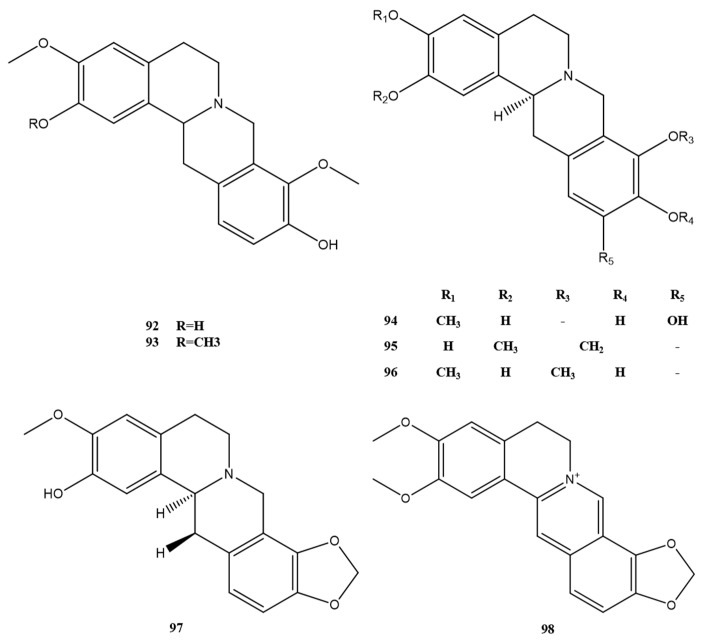
Protopberberine and berberine alkaloids in *M. Rhizoma*.

**Figure 9 molecules-28-02701-f009:**
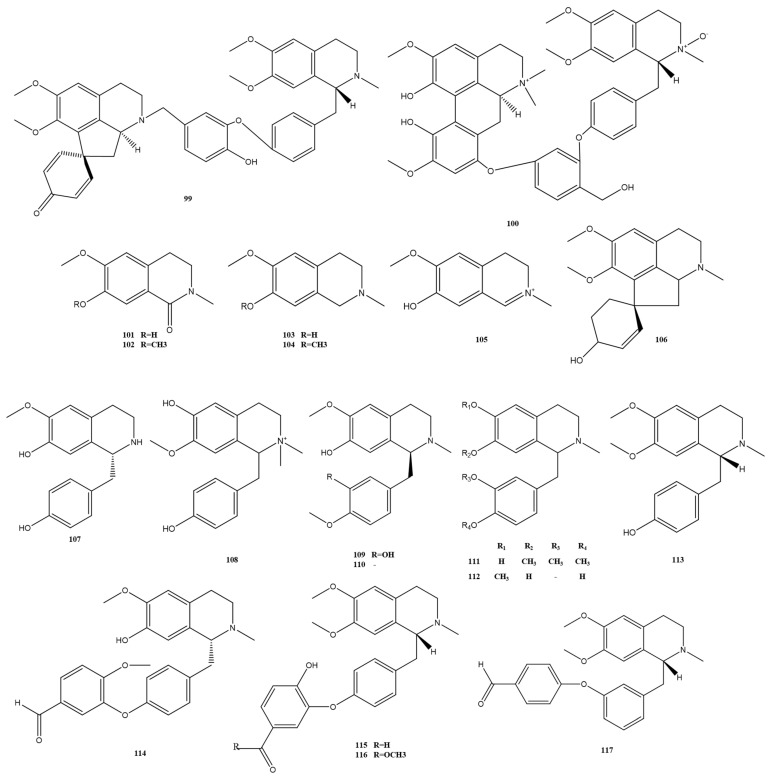
Apomorphine–benzylisoquinoline alkaloids (99–100), simple isoquinoline alkaloids (101–106), and monobenzylisoquinoline alkaloids (107–117) in *M. Rhizoma*.

**Table 1 molecules-28-02701-t001:** Alkaloids of *M. Rhizoma*.

No.	Alkaloids	Formula	Mass	Reference
1	dauricine	C_38_H_44_N_2_O_6_	624.3	[5]
2	daurinoline	C_37_H_42_N_2_O_6_	610.3	[6]
3	dauricinoline	C_37_H_42_N_2_O_6_	610.3	[6]
4	daurisoline	C_37_H_42_N_2_O_6_	610.3	[6]
5	dauricicline	C_36_H_40_N_2_O_6_	596.3	[7]
6	dauricoline	C_36_H_40_N_2_O_6_	596.3	[7]
7	(R,R)-*N*-Desmethlydauricine	C_37_H_42_N_2_O_6_	610.3	[8]
8	*O*-methyldauricine	C_39_H_46_N_2_O_6_	638.3	[7]
9	tetrandrine	C_38_H_42_N_2_O_6_	622.3	[6]
10	thalifortine	C_37_H_40_N_2_O_6_	608.3	[7]
11	costaricine	C_35_H_38_N_2_O_6_	582.3	[9]
12	cycleapeltine	C_37_H_40_N_2_O_6_	608.3	[10]
13	homoaromoline	C_37_H_40_N_2_O_6_	608.3	[10]
14	(+)-1,3,4-dehydrocepharanthine	C_36_H_32_N_2_O_6_	588.2	[11]
15	(+)-1,3,4-dehydrocepharanthine-2′β-*N*-oxide	C_36_H_32_N_2_O_7_	604.2	[11]
16	(1R, 1′R)-dauricine-2β-*N*-oxide	C_37_H_42_N_2_O_7_	626.3	[12]
17	(1R, 1′R)-daurisoline-2β-*N*-oxide	C_38_H_44_N_2_O_7_	640.3	[12]
18	(1R, 1′R)-dauricine-2α-*N*-oxide	C_38_H_44_N_2_O_7_	640.3	[12]
19	(1R, 1ʹR)-dauricisoline A-2α-*N*-oxide	C_38_H_45_N_2_O_7_^+^	641.3	[12]
20	(1R, 1′R)-daurisoline-2′α-*N*-oxide	C_37_H_42_N_2_O_7_	626.3	[12]
21	(1R, 1′R)-dauricine-2′α-*N*-oxide	C_38_H_44_N_2_O_7_	640.3	[12]
22	(1R, 1′R)-dauricisoline C-2′α-*N*-oxide	C_39_H_47_N_2_O_7_^+^	655.3	[12]
23	(1R, 1′R)-dauricine-2′β-*N*-oxide	C_38_H_44_N_2_O_7_	640.3	[12]
24	(1R, 1′R)-dauricisoline E-2′β-*N*-oxide	C_39_H_47_N_2_O_7_^+^	655.3	[12]
25	(1R, 1′R)-dauricisoline A	C_38_H_45_N_2_O_6_^+^	625.3	[12]
26	(1R, 1′R)-dauricisoline B	C_37_H_43_N_2_O_6_^+^	611.3	[12]
27	(1R, 1′R)-dauricisoline C	C_38_H_45_N_2_O_6_^+^	625.3	[12]
28	(1R, 1′R)-dauricisoline D	C_37_H_43_N_2_O_6_^+^	611.3	[12]
29	(1R, 1′R)-dauricisoline E	C_39_H_47_N_2_O_6_^+^	639.3	[12]
30	(1R, 1′R)-espinin	C_36_H_40_N_2_O_6_	596.3	[12]
31	(1′R)-dauricisoline F	C_38_H_43_N_2_O_6_^+^	623.3	[13]
32	(1R, 1′R)-dauricisoline G	C_44_H_48_N_2_O_7_	716.3	[13]
33	(1R, 1′R)-dauricisoline H	C_37_H_43_N_2_O_6_^+^	611.3	[13]
34	(1R, 1′R)-dauricisoline I	C_36_H_40_N_2_O_6_	596.3	[13]
35	(1R, 1′R)-dauricisoline J	C_38_H_46_N_2_O_6_^2+^	626.3	[13]
36	(1′R)-pavermenidaurine	C_38_H_41_N_2_O_7_^+^	637.3	[13]
37	cissampentin	C_37_H_40_N_2_O_6_	608.3	[9]
38	cycleatjehenine	C_37_H_36_N_2_O_6_	604.3	[9]
39	neosutchuenenine	C_36_H_40_N_2_O_6_	596.3	[10]
40	cissampentine A	C_36_H_38_N_2_O_6_	594.3	[11]
41	cissampentine B	C_37_H_40_N_2_O_6_	608.3	[11]
42	(−)-pseudocurine	C_36_H_38_N_2_O_6_	594.3	[11]
43	sutchueneneonine	C_36_H_40_N_2_O_6_	596.3	[10]
44	sutchuenenine	C_36_H_4_0N_2_O_6_	596.3	[10]
45	secoisotetrandrine	C_38_H_40_N_2_O_8_	652.3	[10]
46	tuduranine	C_18_H_19_NO_3_	297.1	[7]
47	iso-corydine	C_20_H_23_NO_4_	341.2	[7]
48	cepharanthine	C_19_H_19_NO_3_	309.1	[7]
49	menisperine	C_21_H_26_NO_4_^+^	356.2	[14]
50	magnoflorine	C_20_H_24_NO_4_^+^	342.2	[14]
51	*N*-formyldehydroanonain	C_18_H_13_NO_3_	291.1	[15]
52	*N*-demethyl-*N*-formyldehydronuciferine	C_19_H_17_NO_3_	307.1	[15]
53	sinotumine G	C_18_H_15_NO_3_	293.1	[16]
54	6-acetyl-5,6-dihydro-1,2-dimethoxy-4H-dibenzo[de,g]-quinoline	C_20_H_19_NO_3_	321.1	[17]
55	*N*-formylmornuciferin	C_19_H_19_NO_3_	309.1	[15]
56	*N*-formylannonaine	C_18_H_15_NO_3_	293.1	[15]
57	*N*-acetylasimilobine	C_19_H_19_NO_3_	309.1	[16]
58	stepharine	C_18_H_19_NO_3_	297.1	[18]
59	telisatin A	C_20_H_15_NO_4_	333.1	[16]
60	telazoline	C_17_H_12_N_2_O_2_	276.1	[11]
61	oxidized nantenine	C_19_H_13_NO_5_	335.1	[7]
62	atherospermidine	C_18_H_11_NO_4_	305.1	[16]
63	dauriporphine	C_20_H_17_NO_5_	351.1	[14]
64	menisporphine	C_19_H_15_NO_4_	321.1	[14]
65	6-*O*-demethylmenisporphine	C_18_H_13_NO_4_	307.1	[14]
66	dauriporphinoline	C_19_H_15_NO_5_	337.1	[14]
67	bianfugecine	C_18_H_13_NO_3_	291.1	[19]
68	bianfugedine	C_18_H_11_NO_4_	305.1	[19]
69	oxoisoaporphine A	C_18_H_11_NO_4_	305.1	[20]
70	oxoisoaporphine B	C_18_H_13_NO_4_	307.1	[20]
71	menisoxoisoaporphine B	C_19_H_15_NO_3_	305.1	[17]
72	menispeimin A	C_17_H_11_NO_3_	277.1	[16]
73	sinotumine D	C_19_H_13_NO_5_	335.1	[16]
74	lakshminine	C_17_H_12_N_2_O_2_	276.1	[11]
75	menisoxoisoaporphine A	C_24_H_26_N_2_O_4_	406.2	[11]
76	daurioxoisoporphine B	C_19_H_16_N_2_O_4_	336.1	[17]
77	Menisoxoisoaporphine C	C_27_H_24_N_2_O_4_	440.2	[17]
78	tyraminoporphine	C_27_H_24_N_2_O_5_	456.2	[11]
79	daurioxoisoporphine A	C_26_H_22_N_2_O_4_	426.2	[7]
80	2,3-dihydrodauriporphine	C_20_H_19_NO_5_	353.1	[7]
81	dihydromenisporphine	C_19_H_17_NO_4_	323.1	[16]
82	sinotumine F	C_18_H_15_NO_6_	341.1	[7]
83	sinomenine	C_19_H_23_NO_4_	329.2	[6]
84	scrodentoside A	C_19_H_23_NO_4_	329.2	[11]
85	disinomenine	C_38_H_44_N_2_O_8_	656.3	[21]
86	dechloroacutumine	C_19_H_25_NO_6_	363.2	[5]
87	dauricumine	C_19_H_24_ClNO_6_	387.1	[5]
88	dauricumidine	C_18_H_22_ClNO_6_	383.1	[14]
89	acutumine	C_19_H_24_ClNO_6_	397.1	[5]
90	acutuminine	C_19_H_24_ClNO_5_	381.1	[14]
91	acutumidine	C_18_H_22_O_6_NCl	383.1	[14]
92	stopholidine	C_19_H_21_NO_4_	327.1	[4]
93	corydalmine	C_20_H_23_NO_4_	341.2	[7]
94	pessoine	C_18_H_19_NO_4_	313.1	[7]
95	cheilanthifoline	C_19_H_19_NO_4_	325.1	[7]
96	stepholidine	C_19_H_21_NO_4_	327.1	[7]
97	(+)-cheilanthifoline	C_19_H_19_NO_4_	325.1	[17]
98	epiberberine	C_20_H_18_NO_4_^+^	336.1	[14]
99	(6aS, 1′R)-apormenidaurine A	C_44_H_46_N_2_O_7_	714.3	[13]
100	(6aS, 1′S)-apormenidaurine B	C_46_H_51_N_2_O_10_	791.4	[13]
101	thalifoline	C_11_H_13_NO_3_	207.1	[7]
102	*N*-methylcorydaldine	C_12_H_15_NO_3_	221.1	[7]
103	corypalline	C_11_H_15_NO_2_	193.1	[7]
104	*O*-methylcorypalline	C_12_H_17_NO_2_	207.1	[7]
105	pycnarrhine	C_11_H_14_NO_2_^+^	192.1	[22]
106	amurolin	C_19_H_25_NO_3_	315.2	[22]
107	coclaurine	C_17_H_19_NO_3_	285.1	[7]
108	lotusine	C_19_H_24_NO_3_^+^	314.2	[7]
109	reticuline	C_19_H_23_NO_4_	329.2	[7]
110	(R)-6-methoxy-1-(4-methoxybenzyl)-2-methyl-1,2,3,4-tetrahydroisoquinolin-7-ol	C_19_H_23_NO_3_	313.2	[7]
111	pseudolaudanine	C_20_H_25_NO_4_	343.2	[7]
112	*N*-methylcoclaurine	C_18_H_21_NO_3_	299.2	[7]
113	armepavine	C_19_H_23_NO_3_	313.2	[7]
114	pecrassipine B	C_26_H_27_NO_5_	433.2	[7]
115	menidaurine A	C_26_H_27_NO_5_	433.2	[23]
116	menidaurine B	C_27_H_29_NO_6_	463.2	[23]
117	menidaurine C	C_26_H_27_NO_5_	433.2	[23]

**Table 2 molecules-28-02701-t002:** Other components in *M. Rhizoma*.

No.	Component	Formula	Mass	Reference
1	p-hydroxyphenethyltrans-ferulate	C_18_H_18_O_5_	314.1	[14]
2	daucosterol	C_35_H_60_O_6_	576.4	[14]
3	vanillin	C_8_H_8_O_3_	152.0	[5]
4	*N*-trans-feruloyltyramine	C_18_H_19_NO_4_	313.1	[5]
5	β-sitostenone	C_30_H_52_O	428.4	[5]
6	β-sitosterol	C_30_H_52_O	428.4	[5]
7	aristoloterpenate I	C_32_H_31_NO_8_	557.2	[5]
8	aristolochic acid	C_17_H_11_NO_7_	341.1	[4]
9	aristolactone	C_15_H_20_O_2_	232.1	[4]
10	eleutheroside d	C_34_H_46_O_18_	742.3	[4]
11	vanillic acid	C_8_H_8_O_4_	168.0	[27]
12	4-hydroxybenzaldehyde	C_7_H_6_O_2_	122.0	[27]
13	syringaldehyde	C_9_H_10_O_4_	182.1	[27]
14	2-hydroxy-1-(4-hydroxy-3,5-dimethoxyphenyl)-1-propanone	C_7_H_6_O_2_	226.1	[27]
15	methyl 4-hydroxyphenylacetate	C_9_H_10_O_3_	166.1	[27]
16	2-(4-hydroxyphenyl)-nitroethane	C_8_H_9_NO_3_	167.1	[27]
17	4-hydroxybenzyl cyanide	C_8_H_7_NO	133.1	[27]
18	dibutyl phthalate	C_16_H_22_O_4_	278.2	[27]
19	fragransin b2	C_11_H_14_O_5_	226.1	[27]
20	7-hydroxy-3,6-dimethoxy-1,4-phenanthraquinone	C_16_H_12_O_5_	284.1	[27]
21	palmitic acid	C_16_H_32_O_2_	256.2	[27]
22	arachidic acid	C_20_H_40_O_2_	312.3	[27]
23	β-stigmasterol	C_29_H_48_O	412.4	[27]
24	ethyl pentamethylbenzene	C_13_H_26_	182.2	[28]
25	tetradecane	C_14_H_30_	198.2	[28]
26	2,6,10-trimethylhexadecane	C_17_H_36_	240.3	[28]
27	octadecane	C_18_H_38_	254.3	[28]
28	diheptadecane	C_27_H_56_	380.4	[28]
29	methyldecanoate	C_11_H_22_O_2_	186.2	[28]
30	2,4-bis(1,1-dimethylethyl-)phenol	C_14_H_28_O	212.2	[28]
31	12-methyl-methyltridecanoate	C_15_H_30_O_2_	242.2	[28]
32	2-dodecen-1-yl(-1)succinic anhydride	C_16_H_25_O_3_	265.2	[28]
33	9-hexadecenoic acid	C_16_H_30_O_2_	254.2	[28]
34	2-methyl-1-hexadecanol	C_17_H_24_O	244.2	[28]
35	7-methyl-tetradecene(z)-1-ol acetate	C_17_H_29_O_2_	265.2	[28]
36	methylpalmitate	C_17_H_34_O_2_	270.3	[28]
37	14-methyl-methylhexadecanoate	C_18_H_36_O_2_	284.3	[28]
38	8,11-methyl-octadecadienoate	C_19_H_32_O_2_	292.2	[28]
39	methyllinoleate	C_19_H_34_O_2_	294.3	[28]
40	methyloleate	C_19_H_36_O_2_	296.3	[28]
41	methylstearate	C_19_H_38_O_2_	298.3	[28]
42	16-methyl-methylheptadecanoate	C_19_H_38_O_2_	298.3	[28]
43	methyleicosanoate	C_21_H_42_O_2_	326.3	[28]
44	20-methyl-methyldocosanoate	C_22_H_44_O_2_	340.3	[28]
45	isopropyl-5,6,19-dioctadecatrienoate	C_31_H_56_O_2_	460.4	[28]
46	2,2,2-trifluoroethyl-9-octadecadienoic acid	C_20_H_33_F_3_O_2_	362.2	[28]
47	2-2-amino-*n*-(3,4,4a,5,6,7-hexahydro-5,6,8-trihydroxy-3-methyl-1-oxo-1h-2-benzopyran-4-yl)-propanamide	C_13_H_20_N_2_O_6_	300.1	[28]
48	3-methylbenzyl alcohol, *tert*-butyldimethylsilyl ether	C_14_H_30_OSi	242.2	[28]
49	hexaethylcyclotrisiloxane	C_12_H_30_O_3_Si_3_	390.1	[28]
50	6,6,8,8,10,10-hexamethyl-2,5,7,9,11,14-hexaoxa-6,8,10-trisilicopentadecane	C_12_H_32_O_6_Si_3_	356.2	[28]
51	octamethylcyclotrisiloxane	C_8_H_24_O_4_Si_4_	296.1	[28]
52	decamethylcyclotrisiloxane	C_10_H_30_O_5_Si_5_	370.1	[28]
53	dodecamethylcyclotrisiloxane	C_12_H_36_O_6_Si_6_	444.1	[28]
54	tetradecamethylcyclotrisiloxane	C_14_H_42_O_7_Si_7_	518.1	[28]
55	hexadecamethylcyclotrisiloxane	C_16_H_48_O_8_Si_8_	592.2	[28]

## Data Availability

No new data were created or analyzed in this study. Data sharing is not applicable to this article.

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
