# Peer review of "Research Progress on Chemical Constituents and Pharmacological Activities of Menispermi Rhizoma"

_molecules, 2023, doi:10.3390/molecules28062701_

Round 1
Reviewer 1 Report
The introduction is very short but it presents the essence of the need to collect the information contained in the manuscript. In my opinion, the literature review was prepared very reliably. 63 monographs on the exact subject were analyzed. The description and tabulation of all the tested ingredients contained in M. Rhizoma is made clear and transparent. The formulas showing the locations of the substituent groups are well represented. Conclusions are correctly formulated and relate directly to the material contained in the manuscript.
The manuscript was prepared very well.
I can only have editorial comments:
l.35 and l.130 M. Rhizoma - It should be written in italic
There should be no space between the digit and the % - l.38, l.101, l.122, l.224
l.173 there should be a space after [45]
l.234 there should be a space after [26]
Author Response
请参阅附件。

Reviewer 2 Report
Research Progress on Chemical constituents and Pharmacological activities of Menispermi Rhizoma
Title: This review is about alkaloids of Menispermi Rhizoma (90%). The title can be updated as “Alkaloids of Menispermi Rhizoma and their pharmacological activities’’. Even most of the pharmacological activities discussed are based/related to alkaloids.
Abstract: Well summarized
1. Introduction
A paragraph about problem statement is missing. Gaps? Why this research? Contribution of this review???
2. Chemical composition
Line 38: 1.7 % - 2.5 %: Put only % on last name: 1.7 - 2.5 %
Line 43. Add a sentence explaining the classification of alkaloids. This classification is based on what?
Table 1. Alkaloids of M. Rhizoma: Arrange the compounds by increasing of their masses (or alphabetical order), and update references accordingly.
Line 89: 2-demethylepiphylline, 5-hydroxylepiphylline and tamsulosin. This sentence is incomplete and no sense. Reformulate or delete it.
L173: [45].Studies have….Put space after full stop.
L214: above 90 %,: No space between number and %, plz correct this mistake in the whole doc
3.4. Anti-bacterial effect
Plz all the names of microorganisms have to be written in italic. Correct this mistake in this section and re-check the entire doc.
L245-249: Another five oxidized…..this sentence is written in poor English and no sense. Try to reformulate starting from: Other five oxidized…..
3.6. Anti-hypoxic effect
The paragraph of this section has 7 lines in one sentence. Plz divide this sentence into 4 small sentences.
Conclusion: ok
References: ok
Reviewer 3 Report
The manuscript entitled «Research Progress on Chemical constituents and Pharmacological activity of Menispermi Rhizoma » is a review study on the rhizome of Menispermum dauricum DC., used in traditional Chinese medicine, for many different ailments. From a well-documented literature data compilation, the paper gathered detailed chemical constituents of M.Rhizoma composed mainly by alkaloids and different class of components (phenolic acids, quinones, cardiotonic glycosides, ..) supposed to be responsible of its many interesting biological activities such as anti-tumour, anti-inflammation, anti-oxidation, bacteriostasis, cardio-cerebrovascular protection, anti-depression and anti-Alzheimer's disease. The manuscript is well written within synthetic tables and well illustrated by drawings of skeleton and structures of main components. But minor revision is required to improve the manuscript. below are a few comments and concerns :
- Lign 51 : correct « Menispermum dauricum » instead of « Menisphermum Dauricum » and should be put in italics
- Lign 224-Lign éé6 : « E. coli, S. aureus, B. subtilis » to be put in italics
- Table 2 (Other compounds) : is the compound 18 (dibutyl phthalate) can be considered as a true natural product compound or a plasticizer contaminant ?
- Table 2 (Other compounds) : the components 48-55 (3-methylbenzyl alcohol-tert-butyldimethylsilyl ether, hexaethylcyclotrisiloxane, 6,6,8,8,10,10-hexamethyl-2,5,7,9,11,14-hexaoxa6,8,10-trisilicopentadecane, octamethylcyclotrisiloxane, decamethylcyclotrisiloxane, dodecamethylcyclotrisiloxane, tetradecamethylcyclotrisiloxane, hexadecamethylcyclotrisiloxane) seemed to be siloxane derivatives of the constituents and may be corrected.
So, I recommend the acceptation of this paper to be published in Molecules journal after these required minor revisions.
